# Changes in Phenylacetylglutamine Levels Provide Add-On Value in Risk Stratification of Hypertensive Patients: A Longitudinal Cohort Study

**DOI:** 10.3390/metabo15010064

**Published:** 2025-01-20

**Authors:** Xuan Xu, Lixin Jia, Bokang Qiao, Yanyan Gong, Shan Gao, Yuan Wang, Jie Du

**Affiliations:** 1Beijing Anzhen Hospital, The Key Laboratory of Remodeling-Related Cardiovascular Diseases, Ministry of Education, Beijing Collaborative Innovation Center for Cardiovascular Disorders, Capital Medical University, No. 2 Anzhen Road, Chaoyang District, Beijing 100029, China; 122022010375@mail.ccmu.edu.cn (X.X.); bokang_qiao@ccmu.edu.cn (B.Q.); gongyanyan03@gmail.com (Y.G.); shangao@mail.ccmu.edu.cn (S.G.); wangyuan980510@ccmu.edu.cn (Y.W.); 2Beijing Institute of Heart, Lung & Blood Vessel Disease, No. 2 Anzhen Road, Chaoyang District, Beijing 100029, China; 3Institute for Biological Therapy, Henan Academy of Innovations in Medical Science, Zhengzhou 450046, China; lixinjia1985@gmail.com

**Keywords:** phenylacetylglutamine (PAGln), hypertension, risk stratification, biomarker, cardiovascular events

## Abstract

Background: Despite antihypertensive treatment, some high-risk hypertensive patients still experience major adverse cardiovascular events (MACEs). Current risk stratification tools may underestimate the presence of metabolites in hypertension and thereby risk of MACEs. Objectives: We aimed to explore the potential value of gut microbiota-derived metabolite phenylacetylglutamine (PAGln) in risk stratification of hypertension. Methods: We measured plasma PAGln levels using liquid chromatography tandem mass spectrometry in 1543 high-risk hypertensive patients, dividing them into a discovery cohort (n = 792) and a validation cohort (n = 751). After follow-up, the Kaplan–Meier curve and the Cox regression model were utilized to determine the correlation between PAGln and MACEs (death, non-fatal ischemic stroke and hemorrhagic stroke, non-fatal acute coronary syndrome and unplanned revascularization). We examined the predictive performance of PAGln in different subgroups and evaluated the incremental predictive value of PAGln as an addition to the ASCVD risk assessment model. Results: Among all high-risk hypertensive patients, 148 patients experienced MACEs after a mean follow-up of 3.02 years. In both cohorts, after adjusting other confounding risk factors, PAGln remained an independent risk factor the MACEs in hypertensive patients. Patients with plasma PAGln ≥ 1.047 μmol/L have a higher risk of MACEs. PAGln concentration provided incremental predictive value to the ASCVD risk model, with better performance in the discovery cohort. It was most effective in female, patients with a systolic blood pressure (SBP) ≥ 130 mmHg and taking angiotensin-converting enzyme inhibitors (ACEIs). Conclusions: PAGln was associated with an increased risk of MACEs in hypertension, especially in women or in subgroups with SBP ≥ 130 mmHg and taking ACEIs. PAGln should be considered as an independent predictor in risk stratification to improve prognosis.

## 1. Introduction

Hypertension is a significant risk factor that contributes to the global disease burden [1], and it is especially critical in relation to cardiovascular disease (CVD) [2]. Guidelines for treating hypertension advocate for antihypertensive therapy in conjunction with cardiovascular risk factor control [3]. Nevertheless, 7.23–12.66% of patients with hypertension still experience cardiovascular events, despite available treatments [3]. In a meta-analysis of 344,716 hypertensive patients from 48 randomized clinical trials around the world, at least 42,324 participants (12.28%) had a major cardiovascular event after a median follow-up of 4.15 years [4]. Another worldwide meta-analysis observed that 9.62% treated hypertensive patients experienced MACEs [5]. This suggests that risk for MACEs cannot be explained completely by the presence of traditional risk scores. High-risk patients with hypertension account for the majority of the hypertensive population [6], and risk stratification needs to be refined to identify individual cardiovascular risks.

Metabolic disorders are involved in the occurrence and development of hypertension [7]. The interaction between genetic and environmental risk factors leads to metabolic disorders [8], which further cause the development of hypertension and target organ damage [7]. This in turn leads to a further increase in metabolic disorder. Metabolic disorders promote increased cardiovascular risk in hypertensive patients independently of blood pressure [9].

Gut microbiota metabolites, including secondary bile acids [10], trimethylamine N-oxide [11], and indoxyl sulfate [12], promote vascular endothelial inflammation [13] and thrombotic potential [14], and they have been associated with an increased risk of CVD. Phenylacetylglutamine (PAGln) is a gut microbiota metabolite derived from the combination of phenylacetic acid (PAA) and glutamine [12]. Existing studies have found that this microbial-derived metabolite is associated with MACEs and acts through adrenergic receptors to increase platelet activity [15]. In a large heart failure cohort, an elevated PAGln level was associated with the primary endpoint event, which indicates that this metabolite can serve as a predictor for the risk of adverse cardiovascular events in patients with heart failure [16]. However, in hypertensive patients, the association between circulating PAGln plasma level and risk of MACEs remains unknown. Therefore, as PAGln is a newly identified risk factor, it is necessary to explore the incremental predictive value of PAGln as an addition to traditional cardiovascular risk assessment models. It is widely recognized that no single risk factor accurately predicts adverse outcomes in all hypertensive patients, and the main subgroups in which PAGln is a predictive factor require identification.

In this study, we investigated the association between plasma PAGln level and the risk of MACEs in a longitudinal cohort of patients with hypertension. We hypothesized that PAGln could be a risk stratification indicator for patients with hypertension.

## 2. Materials and Methods

### 2.1. Study Design and Population

The eligible population was from a Registry Study of Prognostic Risk of Patients with Essential Hypertension for Cardiovascular Events (PROSPECT) that enrolled patients who were admitted to Beijing Anzhen Hospital and the First Hospital of Shanxi Medical University from 2016 to 2021, with principal diagnosis of essential hypertension at discharge. This study was approved and registered by the China Clinical Trial Registration Center, registration number NCT03708601.

Our cohort included 1543 patients with essential hypertension. The discovery cohort (n = 792) consisted of patients with hypertension who came to Beijing Anzhen Hospital between September 2016 and December 2018. For validation of the associations of PAGln and the risk of MACEs in hypertensive patients, 751 patients were enrolled between January 2019 and October 2021 from two hospitals: Beijing Anzhen Hospital (n = 620) and the First Hospital of Shanxi Medical University (n = 131).

The inclusion criteria included an age of 18–85 years and a diagnosis of high-risk hypertension. The diagnostic criteria for high-risk hypertension were any one of the following according to the guidelines [17]: (1) any one of the following diseases, including coronary heart disease (CHD), congestive cardiac failure, atrial fibrillation, cerebrovascular disease or peripheral arterial occlusive disease, and diabetes mellitus, (2) any form of organ damage, which included left ventricular mass index ≥ 115 g/m^2^ in men and ≥95 g/m^2^ in women, carotid intima–media thickness ≥ 0.9 mm or carotid plaque, pulse wave velocity (PWV) ≥ 12 m/s, ankle–brachial index (ABI) < 0.9, serum creatinine ≥ 115 µmol/L in men and ≥107 µmol/L in women, or urine albumin–creatinine ratio ≥ 30 mg/g, (3) grade 1 or 2 hypertension with at least 3 other risk factors including age >55 years in men and >65 years in women, smoking (current or past history), impaired glucose tolerance (fasting blood glucose between 6.1 and 6.9 mmol/L), low-density lipoprotein cholesterol (LDL-C) ≥ 4.1 mmol/L or high-density lipoprotein cholesterol (HDL-C) ≤ 1.0 mmol/L, family history of premature CVD, body mass index (BMI) ≥ 28 kg/m^2^, plasma homocysteine ≥ 15 µmol/L, or (4) grade 3 hypertension.

The exclusion criteria included acute myocardial infarction, unstable angina, or congestive heart failure within the past 3 months; percutaneous transluminal coronary angioplasty or coronary artery bypass grafting within the past 3 months; cerebral infarction, cerebral hemorrhage, or transient ischemic attack within the past 3 months; pregnancy; severe liver disease; renal failure; malignant tumor; and loss to follow-up. Please see Appendix A for detailed inclusion and exclusion criteria. The same inclusion and exclusion criteria were used for the discovery and validation cohorts.

This study complied with the Declaration of Helsinki and was approved by the Medical Ethics Committee of Anzhen Hospital of Capital Medical University. Written informed consent was obtained from all patients.

### 2.2. Plasma PAGln Measurements

Plasma samples were tested in the laboratory. Venous blood was collected, and ethylenediaminetetraacetic acid anticoagulant was added. The samples were then centrifuged and stored at −80 °C.

A volume of 20 μL of serum or PAGln standard working solution was mixed with 20 μL of internal standard solution, followed by the addition of 40 μL of methanol and 120 μL of acetonitrile. The mixture was vortexed and then centrifuged at 14,000 rpm for 10 min at 4 °C. The supernatant (approximately 100 μL) was transferred into a sealed injection vial for analysis [15]. PAGln was obtained from Toronto Research Chemicals (ZC-55087, Toronto, ON, Canada). Isotope standard PAGln-d5 (IR-22231, Toronto Research Chemicals, Toronto, ON, Canada) was used for internal standard solution.

Plasma levels of PAGln were quantified using a targeted liquid chromatography tandem mass spectrometry system (Xevo TQ-S micro, Warters, Milford, MA, USA). PAGln was separated with an ACQUITY UPLC BEH HILIC column (2.1 mm inner diameter × 100 mm with 1.7 μm particles; Waters, Milford, MA, USA). The injection volume of extracted samples was 2 µL. The parameters used for liquid phase separation and mass spectrometry are listed in Appendix A. The concentration gradient of the PAGln calibration curve was 0.2, 0.5, 1, 2, 5, 10, 20, 50, 100 and 200 μmol/L. Quantification was performed using the ratio of analyte to internal standards relative to the PAGln calibration curves obtained by serial dilution of a mixture of PAGln and PAGln-d5.

### 2.3. Outcomes

Follow-up outcomes were MACEs, including death, cerebrovascular events (non-fatal ischemic stroke, and hemorrhagic stroke) and cardiac events (non-fatal acute coronary syndrome, and unplanned revascularization).

### 2.4. Statistical Analyses

Continuous variables are expressed as the medians (interquartile ranges), and they were compared using the Mann–Whitney U test or the Kruskal–Wallis test. Categorical variables are expressed as frequencies and percentages, and they were compared using the chi-square test. We imputed missing covariate data using the predictive mean matching method in the R multivariate imputation by chained equation (mice) package.

Potential clinically useful cut-off value was assessed using receiver operating characteristic (ROC) analysis in the discovery cohort. Patients in two cohorts are stratified according to the optimal cut-off value derived from ROC curve analyses. Kaplan–Meier survival curves were drawn to evaluate the relationship between plasma PAGln level and risk of MACEs, and the log-rank test was used for comparison. The Cox regression method was used to analyze the correlation between PAGln level and risk of MACEs. First, a crude model was constructed to clarify whether there was a relationship between PAGln level and MACEs. Next, a multivariable model was used to correct for other cardiovascular event risk factors. These factors included sex, age, CHD, smoking status, diabetes status, LDL-C, and total cholesterol (TC) (multivariable 1 and 2), as well as the use of angiotensin-converting enzyme inhibitors (ACEIs), β-blockers, calcium channel blockers (CCBs), and angiotensin receptor blockers (ARBs) (multivariable 3). Results are presented as hazard ratios (HRs) and 95% confidence intervals (95% CI), using the low-PAGln group as the reference. The proportional hazards assumption was met in all the Cox proportional hazards regression analyses in both cohort (All *p* values > 0.05).

To evaluate whether PAGln concentration has incremental predictive value when added to existing cardiovascular risk assessment models, we selected the atherosclerotic cardiovascular disease (ASCVD) risk assessment model [18]. The parameters used to compute the ASCVD risk score include sex, age, smoking status, TC, systolic blood pressure (SBP), and diabetes status. Then, we assessed the incremental predictive value of the PAGln level using the *C*-statistic, net reclassification improvement (NRI), and integrated discrimination improvement (IDI).

Next, we conducted subgroup analysis to determine the predictive value of PAGln level in subgroups established on the basis of clinical risk variables (sex, smoking status, diabetes status, and SBP) and use of drugs (ACEIs, β-blockers, and CCBs). The experimental Wald test was employed as an interaction test. Then, we performed a sensitivity analysis among all patients. First, we excluded extreme PAGln concentration values (the highest and lowest 2.5%) to reduce their influence. Then, we examined the relationship between plasma PAGln level and risk of MACEs in patients who experienced cardiac events or cardiovascular and cerebrovascular events. Finally, we tested the robustness of our main findings by excluding patients with missing covariate values.

All statistical analyses were performed using R (Version 4.4.1; Vienna, Austria) and Stata/MP 17.0 (StataCorp, College Station, TX, USA). All tests were two sided, with a significance level of 0.05.

## 3. Results

### 3.1. Baseline Characteristics

Based on admission time, we divided all patients into a discovery cohort (n = 792) and a validation cohort (n = 751). Patients were divided into high and low groups for further analysis based on the optimal cut-off value 1.047 (Appendix A). Table 1 summarized the clinical characteristics of the hypertensive patients in two cohorts. In the discovery cohort, with a median follow-up of 2.2 years, 9.09% of patients with hypertension experienced MACEs, the average age was 50 years (interquartile range [IQR], 39–61), and 60.61% of the patients were male. In the validation cohort, with a median follow-up of 3.5 years, 12.26% of patients with hypertension experienced MACEs, the mean age was 57 (interquartile range [IQR], 47–66) and 59.92% were male. In both cohorts, patients in high-PAGln groups were older, and had a higher prevalence of diabetes, coronary heart disease, and cerebrovascular disease; lower TC, TG and LDL-C levels; more carotid plaque.

### 3.2. Association Between PAGln Concentrations and MACEs in Hypertension

According to targeted detection, plasma PAGln levels were significantly elevated in patients who experienced MACEs (Figure 1a). As observed in the discovery cohort, PAGln levels were significantly elevated in patients with hypertension who experienced MACEs in the validation cohort (Figure 1b).

Then, Kaplan–Meier analysis revealed that the risk of MACEs in hypertensive patients with PAGln ≥ 1.047 was higher than patients with PAGln < 1.047 (Figure 2a). To validate our findings, we performed group analyzes based on cut-off value in the validation cohort. Similarly, the prognosis of high-risk hypertensive patients with plasma PAGln < 1.047 is better than that of high-risk hypertensive patients with plasma PAGln ≥ 1.047 (Figure 2b).

Crude and multivariable Cox regression analyses were used to determine the risk of MACEs in the different PAGln concentration groups. In the discovery cohort (Table 2), taking the low-PAGln group (PAGln < 1.047 μmol/L) as a reference, the HR of patients in the high-PAGln group (PAGln ≥ 1.047 μmol/L) was 3.02 (95% CI, 1.85–4.93). After multi-factor adjustment, higher PAGln concentrations were associated with MACEs, independent of age and sex (_adj_HR, 2.20 [1.33–3.65], *p* = 0.002); CHD, smoking status, diabetes status, LDL, and TC (_adj_HR, 2.19 [1.31–3.66], *p* = 0.003); and ACEIs, β-blockers, CCBs, and ARBs (_adj_HR, 2.32 [1.38–3.89], *p* = 0.001). Furthermore, in the validation cohort (Table 2), the predictive value of PAGln concentration as a predictor of MACEs was consistent with that in the discovery cohort. Taking the low-PAGln group as a reference, the HR of the high-PAGln group was 4.15 (95% CI, 2.33–7.38). After adjustment, a higher level of PAGln was associated with MACEs of age and sex (_adj_HR, 2.21 [1.21–4.05], *p* = 0.010); CHD, smoking status, diabetes status, LDL, and TC (_adj_HR, 2.08 [1.13–3.82], *p* = 0.018); and ACEIs, β-blockers, CCBs, and ARBs (_adj_HR, 2.01 [1.12–3.78], *p* = 0.020). Cox regression analysis performed using PAGln as a continuous variable (Table 2) shows that its influence in determining risk of MACEs remained significant, after adjusting for all clinical risk factors in the discovery cohort (_adj_HR, 1.12 [1.00–1.25], *p* = 0.043) and the validation cohort (_adj_HR, 1.05 [1.02–1.07], *p* < 0.001). For each 1 SD increment of log-transformed plasma PAGln in the fully adjusted model, it was still statistically associated with a 33% risk increase (_adj_HR, 1.33 [1.08–1.63], *p* = 0.006) in the discovery cohort and 69% increase (_adj_HR, 1.69 [1.31–2.18], *p* < 0.001) in the validation cohort.

To test for multicollinearity, we calculated variance inflation factors. The results of the item multicollinearity test showed that VIF < 10 and 1/VIF > 0.1, indicating that there is no multicollinearity problem (Appendix A).

### 3.3. Predictive Performance of PAGln in Hypertensive Patients

We selected the ASCVD risk assessment model [18] as traditional models with which to evaluate the incremental predictive value of PAGln concentration by comparing the traditional models with the new models that include PAGln concentration (Table 3). Including PAGln concentration in the ASCVD risk model increased the *C*-statistic from 0.725 to 0.736 (*p* = 0.001) in the discovery cohort. Additionally, the NRI and IDI were 0.069 (95% CI, 0.005–0.133; *p* = 0.036) and 0.025 (95% CI, 0.002–0.048; *p* = 0.031), respectively. We aimed to replicate our findings in the validation cohort. Including PAGln concentration increased the *C*-statistic from 0.775 to 0.779 (*p* = 0.010). The NRI and IDI were 0.045 (95% CI, 0.010–0.080; *p* = 0.012) and 0.099 (95% CI, 0.011–0.187; *p* = 0.028), respectively. However, there was little increment of *C*-statistic in the validation cohort, and the discovery cohort had better performance for MACEs prediction.

### 3.4. Associations Between PAGln and MACEs in Subgroups and Sensitivity Analyses

Figure 3 shows the subgroup analysis of all patients. There were significant interactions between PAGln concentration and sex (*P*_int_ = 0.020) and between PAGln concentration and SBP (*P*_int_ = 0.023); the association between PAGln concentration and MACEs was especially pronounced in female patients and in patients with an SBP ≥ 130 mmHg. The risk of MACEs was also higher among patients taking ACEIs. PAGln is a known adrenergic receptor agonist; however, there was no interaction between PAGln concentration and β-blocker use (*P*_int_ = 0.070).

In a sensitivity analysis, we excluded extreme values of PAGln concentration and found that the quartile with the highest PAGln concentrations was associated with a higher risk of MACEs (Appendix A). This result remained significant in patients whose outcome events included cardiac events (Appendix A) or cardiovascular and cerebrovascular events (Appendix A). Finally, we excluded patients with missing covariate values (Appendix A), and the result still maintained significance.

## 4. Discussion

This study evaluated the predictive performance of plasma PAGln concentration in high-risk hypertensive patients. Our findings indicate a strong correlation between PAGln levels and the risk of major adverse cardiovascular events (MACEs) after adjusting for traditional risk factors. Specifically, hypertensive patients with PAGln ≥ 1.047 μmol/L exhibited a worse prognosis, and this association was replicated in the validation cohort, suggesting that PAGln can effectively identify high-risk individuals.

Baseline comparisons revealed that patients in the validation cohort had a greater combined history of coronary heart disease compared to the discovery cohort. This discrepancy may explain the higher PAGln levels observed in the validation cohort. Previous studies have established a link between PAGln and CHD, with elevated PAGln levels correlate with cardiometabolic risk factors and an increased risk of CHD [19]. Moreover, among patients with suspected coronary artery disease, plasma PAGln levels were found to be higher in those with obstructive CAD [20]. Untargeted metabolomic study further demonstrated that elevated plasma PAGln levels are associated with an increased risk of future CAD [21].

At the same time, compared with the discovery cohort, patients in the validation cohort had a higher incidence of MACEs, which may be related to these patients taking fewer beta-blockers. Research indicates that β-blockers treatment may mitigate the adverse cardiovascular disease-related phenotypes of PAGln observed at physiological levels [15]. In the previous guidelines, five major drug classes were recommended for the treatment of hypertension: ACEIs, ARBs, β-blockers, CCBs, and diuretics (thiazides and thiazide-like diuretics such as chlortalidone and indapamide) [22]. The combined use of antihypertensive drugs plays a crucial role in the management of hypertension, particularly the combination of ACEIs and CCBs [23]. Furthermore, the combination of ACEIs-ARBs with CCBs, such as amlodipine, is essential in the treatment of CVD [24]. β-blockers, which have been emphasized in recent guidelines by the European Society of Hypertension, are frequently employed to address various clinical conditions associated with hypertension [25]. Consequently, further studies should investigate the relationship between PAGln and β-blocker therapy combined with other antihypertensive drug, particularly in relation to their effects on reducing adverse cardiovascular events in hypertensive patients.

Among hypertensive patients treated with ACEIs, those with elevated PAGln levels were more likely to experience MACEs. While ACEIs are known to protect against cardiovascular events, residual cardiovascular risk remains high even in patients receiving ACEI treatment [26]. This may be related to the pathological effects of PAGln through binding to adrenergic receptors. Studies by Hazen et al. [15] have demonstrated that PAGln significantly affects platelet function, enhancing platelet adhesion to the collagen matrix and promoting platelet aggregation in response to agonists. These effects are mediated through G protein-coupled receptors, including α2A, α2B, and β2-adrenergic receptors (β2-ADR), which initiate cellular responses and increase platelet activity. In macrophages, PAGln causes pathological inflammation and aggravates atherosclerosis through the β2-ADR/cAMP/PKA/NF-κB pathway [27]. In addition, in human aortic endothelial cells, PAGln induces oxidative stress by increasing the levels of NOX2 and NF-κB [28]. Pathophysiological factors, including enhanced platelet activity, Endothelial inflammation and oxidative stress, link hypertension to MACEs [29]. Therefore, we speculate that even if hypertensive patients take ACEIs, elevated PAGln levels may increase the risk of subsequent MACEs by activating adrenergic receptors.

Furthermore, we observed that the association between high PAGln concentrations and MACEs was more pronounced in the female subgroup. Elevated PAGln levels in women may be influenced by hormonal changes and differences in gut microbiota composition. Research revealed that older women have higher PAGln levels than men and younger women, a finding consistent with our findings [30]. It has also been reported that the intake of green tea polyphenols (GTP) decreased PAGln levels in menopausal women [31]. Existing studies have shown that bacterial isolates from three phyla can produce PAA: *Bacteroidetes*, *Firmicutes*, and *Proteobacteria* [32]. In addition, the porA gene of *Clostridium sporogenes* contributes to the production of PAA [15]. PAA than conjugates with glutamine to form phenylacetylglutamine. One study observed an increased abundance of *Firmicutes* in hypertensive women, while that of *Bacteroidetes* and *Clostridium* is significantly decreased in hypertensive men [33]. These sex-based differences in gut microbiota contribute to variations in PAGln levels. PAGln levels in women are associated with higher PWV [34], which is an indicator of arterial stiffness and target organ damage in hypertension. Therefore, elevated PAGln levels in women may also contribute to MACEs by affecting arterial stiffness.

Additionally, in the subgroup of patients with an SBP ≥ 130 mmHg, an elevated PAGln concentration was associated with a higher risk of MACEs. Sustained increases in SBP can damage blood vessels and accelerate their stiffening [35]. In the clinical prevention and treatment guidelines for hypertension, PWV is considered a risk factor, as it is involved in target organ damage in patients with hypertension. In existing studies, PAGln concentration has been found to be positively correlated with carotid femoral PWV in renal transplant patients and is a potential risk factor for arterial stiffness in these patients [36]. In addition, PAGln concentration was found to be closely associated with PWV and hypertension in a metabolomics study [34]. Arterial stiffness further causes blood vessel wall thickening and lumen stenosis [37], aggravating hypertension and subsequent MACEs.

Currently, the parameters included in CVD risk assessments are sex, age, TC and diabetes status, all of which have been demonstrated by numerous clinical trials to be independent predictors of CVD. In our cohort, we assessed the incremental predictive value of PAGln concentration when added to the traditional CVD risk factors in the ASCVD risk model [18]. Our findings demonstrated that PAGln concentration provided significant incremental predictive value, suggesting that PAGln can, to some extent, enhance the sensitivity of models used to predict hypertension prognosis.

Based on our longitudinal cohort study of hypertension, we have clarified the relationship between the gut microbiota metabolite PAGln and major adverse cardiovascular events (MACEs) in hypertensive patients. PAGln demonstrated significant incremental value in risk prediction across cardiovascular risk scores. Notably, we observed substantial predictive performance of PAGln in specific subgroups of hypertensive patients. In particular, women, patients with SBP ≥ 130 mmHg, and those taking ACEIs should prioritize PAGln detection. As emerging risk factors, both gut microbiota and PAGln warrant further investigation as potential targets for intervention in the tertiary prevention of hypertension.

This study had several limitations. Our study established an association between PAGln concentration and MACEs among patients with hypertension but did not clearly demonstrate a causal relationship. In addition, this study was limited to the Asian population, had a small sample size, and tracked few follow-up outcomes. Research involving a large multicenter cohort is still needed. Another limitation was that our institution did not routinely assess patient compliance with treatment. Therefore, we recommend closer medical follow-up of high-risk hypertensive patients. Finally, the detailed mechanisms induced by PAGln need further investigation.

## 5. Conclusions

In summary, PAGln concentration is independently associated with MACEs in patients with hypertension, especially in women, high-risk hypertensive patients with SBP ≥ 130 mmHg and patients taking ACEIs. PAGln as a new cardiovascular risk predictor can provide incremental value in the cardiovascular risk stratification of hypertensive patients.

## Figures and Tables

**Figure 1 metabolites-15-00064-f001:**
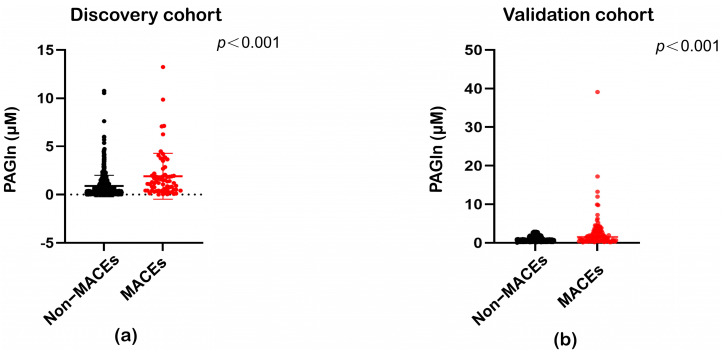
Comparison of plasma phenylacetylglutamine (PAGln) levels among patients with MACEs and non-MACEs. (**a**) Scatter plot shows circulating PAGln concentration in patients with hypertension who experienced MACEs compared with non-MACEs in the discovery cohort. (**b**) Scatter plot shows circulating PAGln concentration in patients with hypertension who experienced MACEs compared with non-MACEs in the validation cohort. The filled red circles represent patients with MACEs and the filled black circles represent patients with non-MACEs. The central line represents the median, the rectangles give the interval between the first and third quartiles. *p* Values were calculated using Mann–Whitney U test.

**Figure 2 metabolites-15-00064-f002:**
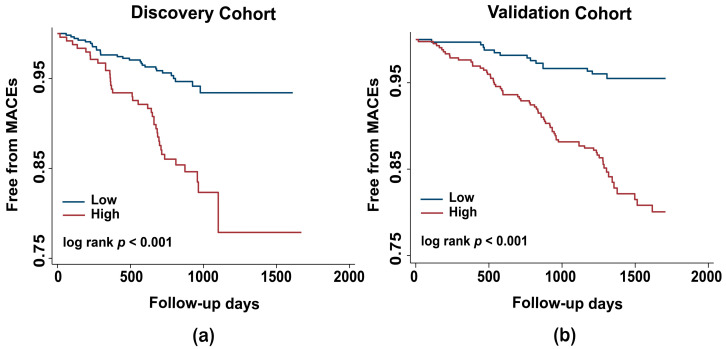
Association between plasma phenylacetylglutamine (PAGln) concentration and risk of MACEs in patients with hypertension. (**a**) Kaplan–Meier estimates of the risk of MACEs in the discovery cohort. (**b**) Kaplan–Meier estimates of the risk of MACEs in the validation cohort. Divided patients into low (<1.047) and high groups (≥1.047) based on cut-off value.

**Figure 3 metabolites-15-00064-f003:**
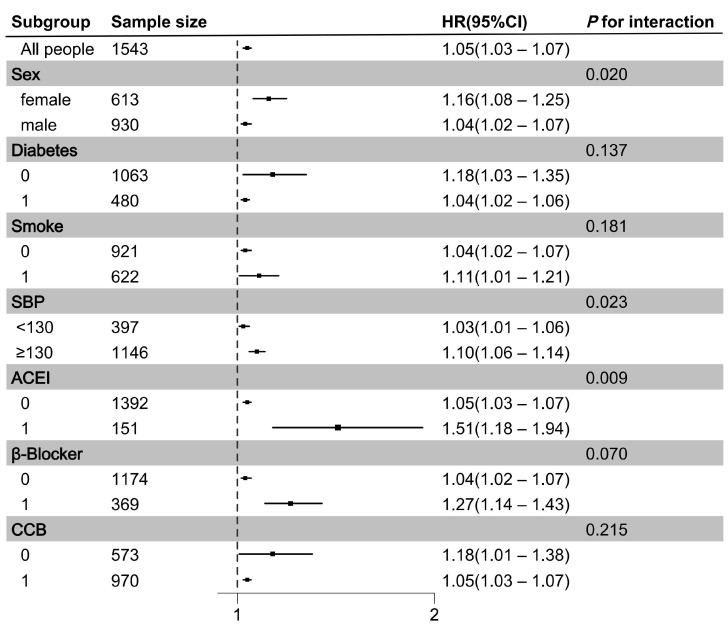
Risk of MACEs according to plasma PAGln concentration (μmol/L) among subsets of patients classified by cardiovascular event risk factors and use of antihypertensive drugs. We included all hypertensive patients and performed subgroup analyses with PAGln concentration as a continuous variable. Hazard ratios (HRs), 95% CIs and *p* values were estimated by multivariate Cox regression and adjusted for age and sex. Abbreviations: SBP, systolic blood pressure; ACEI, angiotensin-converting enzyme inhibitor; CCB, calcium channel blocker.

**Table 1 metabolites-15-00064-t001:** Baseline clinical characteristics for the discovery and validation cohorts ^1^.

	Discovery Cohort				Validation Cohort			
	All (n = 792)	Low ^2^ (n = 548)	High ^2^ (n = 244)	*p* Value	All (n = 751)	Low ^2^ (n = 328)	High ^2^ (n = 423)	*p* Value
Demographic characteristics								
Age (years)	50 [39, 61]	47 [37, 59]	57 [45, 67]	<0.001	57 [47, 66]	51 [40, 60]	62 [52, 69]	<0.001
Sex (male)	480 (60.61%)	347 (63.32%)	133 (54.51%)	0.019	450 (59.92%)	199 (60.67%)	251 (59.34%)	0.712
Smoking	330 (41.67%)	226 (41.24%)	104 (42.62%)	0.716	292 (38.88%)	129 (39.33%)	163 (38.53%)	0.825
Systolic pressure (mmHg)	140 [130, 154]	140 [130, 154]	140 [129, 153]	0.488	139 [128, 151]	141 [130, 154]	138 [127, 150]	0.022
Heart rate (bpm)	71 [64, 80]	71 [64, 80]	69 [62, 79]	0.012	75 [68, 82]	75 [68, 82]	75 [68, 82]	0.926
Medical history								
History of diabetes	240 (30.30%)	149 (27.19%)	91 (37.30%)	0.004	240 (31.96%)	83 (25.30%)	157 (37.12%)	0.001
History of CHD	70 (8.84%)	32 (7.16%)	38 (27.27%)	<0.001	181 (24.10%)	54 (16.46%)	127 (30.02%)	<0.001
History of TIA	104 (13.13%)	63 (11.50%)	41 (16.80%)	0.041	102 (13.58%)	26 (7.93%)	76 (17.97%)	<0.001
Laboratory measurements								
TC (mmol/L)	4.73 [4.07, 5.43]	4.83 [4.20, 5.47]	4.52 [3.75, 5.34]	<0.001	4.57 [3.80, 5.34]	4.79 [3.96, 5.48]	4.37 [3.59, 5.23]	<0.001
TG (mmol/L)	1.51 [1.04, 2.20]	1.57 [1.10, 2.25]	1.36 [0.94, 2.07]	0.007	1.84 [1.25, 3.11]	1.95 [1.32, 3.39]	1.78 [1.15, 2.76]	0.004
HDL cholesterol (mmol/L)	1.16 [1.01, 1.36]	1.15 [1.02, 1.36]	1.16 [0.98, 1.36]	0.781	1.13 [0.95, 1.32]	1.13 [0.96, 1.29]	1.12 [0.94, 1.34]	0.857
LDL cholesterol (mmol/L)	2.91 [2.25, 3.47]	2.96 [2.36, 3.49]	2.66 [2.05, 3.43]	0.005	2.76 [2.13, 3.38]	2.93 [2.32, 3.55]	2.62 [2.00, 3.22]	<0.001
HCY (μmol/L)	11.40 [9.00, 15.70]	11.10 [9.15, 16.35]	12.15 [9.15, 16.35]	0.307	13.20 [11.00, 16.50]	13.15 [10.90, 15.85]	13.20 [11.00, 17.00]	0.588
Cr (μmol/L)	68.90 [57.45, 80.30]	68.10 [56.85, 78.70]	71.30 [59.30, 85.05]	0.005	70.90 [60.50, 83.90]	70.45 [60.90, 84.80]	71.20 [59.70, 82.80]	0.663
Fasting blood glucose (mmol/L)	5.60 [5.13, 6.52]	5.58 [5.13, 6.41]	5.62 [5.12, 6.86]	0.206	5.67 [5.04, 6.99]	5.63 [5.04, 6.63]	5.72 [5.05, 7.12]	0.160
Echocardiography								
EF (%)	65 [62, 68]	65 [62, 68]	65 [62, 68]	0.927	65 [62, 68]	65 [62, 68]	65 [62, 68]	0.944
PWV (m/s)	15.19 [13.92, 17.25]	15.02 [13.82, 16.99]	15.76 [14.20, 17.89]	0.003	14.94 [13.65, 17.11]	14.93 [13.67, 16.90]	15.00 [13.58, 17.21]	0.685
ABI	1.16 [1.11, 1.20]	1.16 [1.11, 1.20]	1.16 [1.10, 1.20]	0.448	1.18 [1.12, 1.24]	1.18 [1.12, 1.25]	1.18 [1.12, 1.24]	0.369
Carotid plaque ^3^	577 (72.85%)	375 (68.43%)	202 (82.79%)	<0.001	477 (63.52%)	175 (53.35%)	302 (71.39%)	<0.001
Medication								
β-blocker	254 (32.07%)	170 (31.02%)	84 (34.43%)	0.343	115 (15.31%)	48 (14.63%)	67 (15.84%)	0.649
ACEI	82 (10.35%)	53 (9.67%)	29 (11.89%)	0.041	69 (9.19%)	34 (10.37%)	35 (8.27%)	0.325
ARB	365 (46.09%)	260 (47.45%)	105 (43.03%)	0.250	257 (34.22%)	105 (32.01%)	152 (35.93%)	0.261
CCB	543 (68.56%)	363 (66.24%)	180 (73.77%)	0.035	427 (56.86%)	194 (59.15%)	233 (55.08%)	0.265
Diuretic	155 (19.57%)	102 (18.61%)	53 (21.72%)	0.482	80 (10.65%)	33 (10.06%)	47 (11.11%)	0.644

^1^ Values for continuous and categorical variables are expressed as the median [25th, 75th percentile] and percentage, respectively. The Kruskal–Wallis test for continuous variables and the chi-square test for categorical variables were used to determine the significant difference between groups. ^2^ PAGln < 1.047 was defined as the low-PAGln group, PAGln ≥ 1.047 was defined as the high-PAGln group. ^3^ Carotid plaque: The presence of a carotid plaque can be defined by an intima–media thickness (IMT) ≥ 1.5 mm, or by a focal increase in thickness of 0.5 mm or 50% of the surrounding carotid IMT value. Abbreviations: CHD, coronary heart disease; TIA, transient ischemic attack; TC, total cholesterol; TG, triglyceride; HDL, high-density lipoprotein; LDL, low-density lipoprotein; HCY, homocysteine; Cr, creatinine; EF, ejection fraction; PWV, pulse wave velocity; ABI, ankle brachial index; ACEI, angiotensin-converting enzyme inhibitor; ARB, angiotensin receptor blocker; CCB, calcium channel blocker.

**Table 2 metabolites-15-00064-t002:** Multivariate-adjusted hazard ratios (HRs) for MACEs associated with plasma concentrations of phenylacetylglutamine (PAGln).

Plasma PAGln (μmol/L)	Number of Events (%)	Crude Model		Multivariable 1 ^1^		Multivariable 2 ^2^		Multivariable 3 ^3^	
		HR [95% CI]	*p* Value	HR [95% CI]	*p* Value	HR [95% CI]	*p* Value	HR [95% CI]	*p* Value
Discovery cohort									
PAGln < 1.047	69.19%	1.0 [referent]		1.0 [referent]		1.0 [referent]		1.0 [referent]	
PAGln ≥ 1.047	30.81%	3.02 [1.85–4.93]	<0.001	2.20 [1.33–3.65]	0.002	2.19 [1.31–3.66]	0.003	2.32 [1.38–3.89]	0.001
PAGln ^4^ (continuous variable)	9.09%	1.29 [1.18–1.41]	<0.001	1.17 [1.06–1.28]	0.030	1.13 [1.01–1.25]	0.033	1.12 [1.00–1.25]	0.043
Per 1-SD ^5^		1.59 [1.29–1.97]	<0.001	1.34 [1.09–1.65]	0.005	1.30 [1.06–1.60]	0.012	1.33 [1.08–1.63]	0.006
Validation cohort									
PAGln < 1.047	43.68%	1.0 [referent]		1.0 [referent]		1.0 [referent]		1.0 [referent]	
PAGln ≥ 1.047	56.32%	4.15 [2.33–7.38]	<0.001	2.21 [1.21–4.05]	0.010	2.08 [1.13–3.82]	0.018	2.01 [1.12–3.78]	0.020
PAGln ^4^ (continuous variable)	12.26%	1.06 [1.05–1.08]	<0.001	1.05 [1.03–1.08]	<0.001	1.05 [1.02–1.07]	<0.001	1.05 [1.02–1.07]	<0.001
Per 1-SD ^5^		2.39 [1.90–3.00]	<0.001	1.85 [1.43–2.40]	<0.001	1.79 [1.33–2.23]	<0.001	1.69 [1.31–2.18]	<0.001

^1^ Multivariable 1 was adjusted for sex and age. ^2^ Multivariable 2 was adjusted for sex, age, CHD, smoking status, diabetes, LDL-C, and TC. ^3^ Multivariable 3 was adjusted for sex, age, CHD, smoking status, diabetes, LDL-C, TC, ACEI, β-blocker, CCB, ARB. ^4^ In Cox regression analysis PAGln as a continuous variable. ^5^ Log-transformed hazard ratios (HRs) and 95% CIs were estimated by plasma PAGln levels. Abbreviations: CHD, coronary heart disease; LDL-C, low-density lipoprotein cholesterol; ACEI, angiotensin-converting enzyme inhibitor; ARB, angiotensin receptor blocker; CCB, calcium channel blocker.

**Table 3 metabolites-15-00064-t003:** Incremental predictive value for MACEs obtained by the addition of PAGln concentration to the ASCVD in the discovery and validation cohorts.

	*C*-Statistic (95% CI)	*p* Value	NRI (95% CI)	NRI *p* Value	IDI (95% CI)	IDI *p* Value
Discovery cohort						
ASCVD ^1^	0.725 [0.665–0.784]		reference		reference	
ASCVD + PAGln	0.736 [0.674–0.797]	0.001	0.069 [0.005–0.133]	0.036	0.025 [0.002–0.048]	0.031
Validation cohort						
ASCVD	0.775 [0.726–0.824]		reference		reference	
ASCVD + PAGln	0.779 [0.730–0.828]	0.010	0.045 [0.010–0.080]	0.012	0.099 [0.011–0.187]	0.028

^1^ ASCVD: cardiovascular risk factors in the ASCVD risk model include sex, age, smoking status, total cholesterol, systolic blood pressure, and diabetes status. Abbreviations: NRI, net reclassification improvement; IDI, integrated discrimination improvement; PAGln, phenylacetylglutamine; ASCVD, atherosclerotic cardiovascular disease. *C*-statistic, NRI, IDI, 95% confidence intervals (CI) and *p* values are results from the multivariable logistic regression model.

## Data Availability

Data are available upon reasonable request to the corresponding authors. The data are not publicly available due to privacy.

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
