# Peer review of "Changes in Phenylacetylglutamine Levels Provide Add-On Value in Risk Stratification of Hypertensive Patients: A Longitudinal Cohort Study"

_metabolites, 2025, doi:10.3390/metabo15010064_

Round 1
Reviewer 1 Report
Comments and Suggestions for Authors
The article creates a positive impression, is well-written with a strong methodological foundation, and highlights both scientific innovation and practical relevance.
Recommendations to Authors.
To strengthen the analysis, test the proportional hazards assumption in the Cox regression and consider alternative methods like time-varying covariates if violated. Address potential multicollinearity among covariates using diagnostics like the variance inflation factor (VIF), and consider analyzing PAGln levels as a continuous variable or with splines for dose-response insights.
If you cannot include these recommendations in the article materials, please indicate them in the limitations section of the study.
Reviewer 2 Report
Comments and Suggestions for Authors
In the presented for peer-reviews original study: "Changes phenylacetylglutamine level provide add-on value on risk stratification in hypertensive patients: a longitudinal cohort study", the Authors have studied the associations between phenylacetylglutamine (PAGln) plasma levels and mace. They have included a cohort of 1543 patients who had hypertension. Patients were divided into model building group (n=792) and verification group (n=751). Patients had PAGln assessment at baseline, and they were followed for a median 2.2 and 3.5 years for MACE.
The idea of this study is interesting and worth of publication. However, before the paper merits publication, some corrections should be made.
1. Abstract. The Authors wrote that MACE was defined as death, non-fatal cerebral ischemia and non-fatal ischemic heart disease. However, in outcomes MACE was defined differently: including hemorrhagic stroke, PCI and unstable angina. Either in abstract and text the outcomes should be defined the same way.
2. Abstract and Results. The Authors found little additive value of PAGln levels when the validation group was analysed in C-statistics (from 0.775 in ASCVD to 0.779 in ASCVD+PAGln). Although, building model had better performance for MACE prediction. This should be clearly stated. The other confusing parameter is a follow-up period of 2.2 years in a model, and 3.5 years in verification.
3. Table 1 should contain baseline characteristics of patients included in primary model and verification model, including age, gender, atherosclerosis risk factors, comorbidities, medications, along with p-values for groups comparisons. This would enable to assess groups homogeneity.
4. Table 1- ABI is reported with errors. The median value should be around 1.0. Values exceeding 2.0 and 3.0 are mistake. PWV report as m/s not as cm/s. Definition of carotid plaque should be included. Blood lowering medications: they do not adhere to guidelines. In the presented age category of patients, initial treatment should include either ACEI+CCB or ARB+CCB, adding diuretic(s) and beta-blocker if required. It seems that patients were undertreated. No data on diuretics and flozins are included. Please report current blood lowering medications in Discussion. The Authors might find utile the following paper to discuss: https://doi.org/10.3390/jcm13051508
5. The cut-off for PAGln should be better explained, and optimally ROC derived.
6. Results are difficult to understand in some parts. Also, I do not understand why PAGln is divided into quartiles, particularly that Quartiles differ between the groups. Cut-off for PAGln and ranges should be recalculated 7. This is somehow strange that Q1, Q2, Q3 and Q4 values are twice as high in validation group compared to a Discovery group.
7. Discussion should be a logical consequence of the results. I have an impression that it is a mixture of loose sentences. Please link your results with other studies on PAGln and MACE.
8. Figure 4. Risk of MACEs according to plasma phenylacetylglutamine (PAGln) concentration among subsets of patients classified... Please explain what PAGln level you mean. There is no such thing in Figure 4.
9. In conclusions, a discovery primary model should propose PAGln cut-offs for MACE prediction deriving from some follow-up period, whereas a validation model should check whether the proposed PAGln cut-off has additive prognostic value for MACE in the same predefined follow-up period.
Comments on the Quality of English LanguageEnglish is sometimes difficult to understand
Reviewer 3 Report
Comments and Suggestions for Authors
I find this article entitled “Changes phenylacetylglutamine level provide add-on value on risk stratification in hypertensive patients: a longitudinal cohort study”, very interesting. I consider that it is well written and presents a thorough and well-founded statistical analysis. However, some aspects need to be improved. My observations and comments are as follows:
Authors should indicate in the Introduction of the article whether the following data corresponds to the worldwide level: “... 7.23 - 12.66% of patients with hypertension still experience cardiovascular events, ...”
The wording of the objective of the abstract does not mention the risk factor for the development of major adverse cardiovascular events (MACEs). At the end of the introduction the objective of the paper is presented and I think it is well written.
Line 77: The authors do not point out the meaning of this note: “(NCT03708601)”.
Line 78: I think the value of “1453” is incorrect, according to my criteria the number should be “1543”. Please revise this value.
Somewhere in the Materials and Methods section, the authors should clearly specify what is meant by discovery cohort and validation cohort, and should mention whether different inclusion criteria were used for the distribution of patients between these two groups. On the other hand, was any specific methodology for validation processes used in the validation cohort?
Line 82-83: The authors present the following sentence: “The inclusion criteria included an age of 18-85 years and a diagnosis of high-risk hypertension”. What does the diagnosis of high-risk hypertension refer to (do they mean high risk for any type of adverse event?).
Line 96: The authors should correct the temperature used for storage of blood samples. It should be -80 °C.
The reaction in section 2.2. Plasma PAGln measurements should be improved. The authors should be more precise in describing the methodology for PAGIn extraction. In addition, they do not describe what the term “PAGIn-d5” means, nor do they mention how they obtained it.
Line 118: in some parts of the article they use the term “Cox” and in other sections they use the term “COX”. The authors should be consistent with the use of this term throughout the paper.
In the Note to Table 2 the authors should include the definition of the abbreviations used in the Table, as well as mention the name of the statistical tests used to calculate the p value.
In the legend of Figure 4 the authors should mention the meaning of the abbreviations used, as well as the name of the statistical tests used to calculate the p value.
Lines 286 and 287: the authors state the following: “It has also been reported that the intake of green tea polyphenols is likely to affect PAGln levels in menopausal women”. However, the authors should indicate in this paragraph how PAGln levels are affected (are they increased or decreased?).
Round 2
Reviewer 2 Report
Comments and Suggestions for Authors
The Authors answered all comments in satisfactory way resulting in the improvement of the paper. I have no further comments. I would like to congratulate Authors on interesting study.